# Biomarkers for Non-Invasive Stratification of Coronary Artery Disease and Prognostic Impact on Long-Term Survival in Patients with Stable Coronary Heart Disease

**DOI:** 10.3390/nu14163433

**Published:** 2022-08-20

**Authors:** Jeffrey Netto, Andrej Teren, Ralph Burkhardt, Anja Willenberg, Frank Beutner, Sylvia Henger, Gerhard Schuler, Holger Thiele, Berend Isermann, Joachim Thiery, Markus Scholz, Thorsten Kaiser

**Affiliations:** 1Leipzig Research Center for Civilization Diseases (LIFE), University of Leipzig, 04109 Leipzig, Germany; 2Institute of Laboratory Medicine, Clinical Chemistry and Molecular Diagnostics, University of Leipzig, 04103 Leipzig, Germany; 3Department of Cardiology and Intensive Care Medicine, University Hospital OWL, Campus Klinikum Bielefeld, 33604 Bielefeld, Germany; 4Institute of Clinical Chemistry and Laboratory Medicine, University Hospital Regensburg, 93053 Regensburg, Germany; 5Department of Internal Medicine/Cardiology, Heart Center Leipzig at University of Leipzig, 04109 Leipzig, Germany; 6Institute for Medical Informatics, Statistics and Epidemiology, University of Leipzig, 04109 Leipzig, Germany; 7Faculty of Medical, Christian-Albrechts-Unversity Kiel, 24105 Kiel, Germany; 8Institute for Laboratory Medicine, Microbiology and Pathobiochemistry, University Hospital Ostwestfalen-Lippe (UK-OWL), Campus Klinikum Lippe, Bielefeld University, 33615 Bielefeld, Germany

**Keywords:** biomarkers, stable coronary artery disease (CAD), troponin T, long-term survival

## Abstract

Knowledge about cardiac and inflammatory biomarkers in patients with stable coronary artery disease (CAD) is limited. To address this, we analyzed 3072 patients (36% female) with a median follow-up of 10 years in the Leipzig LIFE Heart Study with suspected CAD with coronary angiography. Selected biomarkers included troponin T (hsTNT), *N*-terminal pro B-type natriuretic peptide (NT-proBNP), copeptin, *C*-reactive protein (hsCRP), and interleukin-6 (IL-6). Patients were stratified by CAD severity: CAD0 (no sclerosis), CAD1 (non-obstructive, i.e., stenosis < 50%), and CAD2 (≥one stenosis ≥ 50%). Group comparison (GC) included GC1: CAD0 + 1 vs. CAD2; GC2: CAD0 vs. CAD1 + 2. CAD0, CAD1, and CAD2 were apparent in 1271, 631, and 1170 patients, respectively. Adjusted for classical risk factors, hs-cTnT, NT-proBNP, and IL-6 differed significantly in both GC and hsCRP only in GC2. After multivariate analysis, hs-cTnT, NT-proBNP, and IL-6 remained significant in GC1. In GC2, hs-cTnT (*p* < 0.001) and copeptin (*p* = 0.014) reached significance. Ten-year survival in groups CAD0, CAD1, and CAD2 was 88.3%, 77.3%, and 72.4%. Incorporation of hs-cTnT, NT-proBNP, copeptin, and IL-6 improved risk prediction (*p* < 0.001). The studied cardiac and inflammatory biomarkers enable fast and precise non-invasive identification of mortality risk in CAD patients, allowing the tailoring of primary and secondary CAD prevention.

## 1. Introduction

Coronary artery disease (CAD) is a major cause of premature death and disability worldwide [1]. A substantial part of cardiovascular risk is determined by traditional risk factors in both the primary and secondary prevention of CAD [2]. The leading causes for CAD are elevated LDL-cholesterol, followed by hypertension, smoking, diabetes, low HDL-cholesterol, a family history of CAD, and other lifestyle-related factors, such as dietary habits [2]. Considering the complex metabolic, inflammatory, and cellular network of atherosclerotic processes, the precise prediction of an individual’s cardiovascular risk and the mortality risks of CAD patients remains challenging.

To explain the residual variance in cardiovascular risk, several established and novel biomarkers have been evaluated. For example, several markers of inflammation have been associated with the progression and clinical severity of CAD [3]. The Canakinumab Anti-Inflammatory Thrombosis Outcomes Study (CANTOS) [4], with antibodies targeting interleukin-1ß in CAD patients, could consistently demonstrate that lowering the inflammatory burden in CAD patients can reduce the numbers of nonfatal myocardial infarctions, nonfatal strokes, and cardiovascular deaths without changing LDL–cholesterol levels [4]. The benefit of lowering chronic inflammation in CAD patients was supported by the recent Low Dose Colchicine for Secondary Prevention of Cardiovascular Disease Study (LoDoCo) [5], showing the efficacy and safety of low-dose colchicine for secondary prevention in patients with stable CAD. Furthermore, several molecular markers of cardiac damage and cardiovascular stress have emerged in the prediction of adverse events and prognoses in patients with acute coronary syndrome over the past decade [6,7].

However, to date, the roles of these markers in indicating CAD severity and long-term survival (10 years) in a large cohort of angiographically assessed patients with various degrees of cardiovascular risk have not been studied in detail.

The aim of our present study was to investigate in patients of the LIFE-Heart study [8,9,10,11] the associations of five promising, standardized, and robust cardiovascular and inflammatory biomarkers (copeptin, hsTNT, NT-proBNP, hsCRP, and interleukin-6) for non-invasive identification of CAD patients, and mortality risk stratification to enable more precisely primary and secondary cardiovascular prevention.

## 2. Materials and Methods

All analyses were performed for participants in LIFE-Heart, a longitudinal study of patients with suspected or confirmed CAD treated at the Heart Center Leipzig. The study was primarily initiated to identify genetic, biochemical, and environmental markers related to the development of coronary and extra coronary atherosclerosis. A detailed description of the LIFE-Heart study and baseline sample characteristics has been published elsewhere [8]. The study met the ethical standards of the Declaration of Helsinki and was approved by the Ethics Committee of the Medical Faculty of the University of Leipzig, Germany (registration number 276-2005) and is registered with ClinicalTrials.gov (NCT00497887). Written informed consent was obtained from all participants enrolled in the study.

### 2.1. Study Population

We analyzed serum samples of 3305 patients (2122 males and 1183 females) admitted for clinically suspected chronic coronary syndrome, i.e., CAD, who were undergoing first-time coronary angiography. Exclusion criteria included acute myocardial infarction; pregnancy; breast-feeding; and severe systemic diseases, such as autoimmune diseases treated with immune-modulating therapies, diseases requiring dialysis, acute or chronic infectious diseases, and cancer or cancer therapy within the last two years. Upon the exclusion of 233 patients (of which 121 were excluded for unknown angiographic findings and 112 for hs-cTnT ≥ 52 pg/mL; this cut off is based on an assay-specific algorithm (Roche–Elecsys) for myocardial injury [2]), 3072 patients were considered eligible for the present analysis (males *n* = 1966, 64%; and females *n* = 1106, 36%). The age range was 25–87 years (62.2 ± 10.6). Coronary angiography was performed on all participants, as indicated by clinical symptoms or positive non-invasive testing (e.g., cardiopulmonary exercise testing, echocardiography, and nuclear and magnetic resonance imaging). The median follow up-time for survival analysis was 10 years (range 0–14). The flow chart of the study design is shown in Appendix A.

### 2.2. Assessment of Traditional Cardiovascular Risk Factors and Coronary Status

The established risk factors for atherosclerosis were evaluated using a standardized interview, standardized biometric assessment, and laboratory analysis, as described elsewhere [8]. Frequent comorbidities were considered in the evaluation as confounding factors: diabetes mellitus, obesity, hyperlipidemia, hypertension, and impaired renal function. These comorbidities have particular impacts on metabolic parameters, inflammatory markers, hormonal regulations, and the half-lives of circulating biomarkers (e.g., renal function). Other comorbidities were documented and are the subjects of ongoing investigations.

Medications were documented according to the ATC-Codes. For all participants, coronary angiography was performed by following in-house standards. According to the angiographic findings, the participants were divided into the following groups: coronary artery disease was defined by coronary angiography as CAD0 (no coronary sclerosis), CAD1 (non-obstructive coronary sclerosis, i.e., plaques with <50% luminal reduction), or obstructive CAD2 (≥one stenosis ≥ 50% luminal reduction).

### 2.3. Laboratory Analysis

Venous blood was collected before the intervention. Samples were transferred at 4 °C within five hours to our laboratory and immediately centrifuged (except for blood counts), analyzed, and had aliquots frozen in the gas phase of liquid nitrogen. All laboratory procedures were performed according to the accreditation norms ISO 15180 and ISO 17025.

Initial laboratory analysis of the LIFE-Heart study included low-density lipoprotein cholesterol (LDL-C), high-density lipoprotein cholesterol (HDL-C), triglycerides (TG), and creatinine [8].

We selected promising biomarkers for the early and sensitive detection of patients at risk for CAD. These additional biomarkers represent robust cardiac and highly sensitive inflammatory biomarkers that are rapidly available in our clinical laboratory.

Measurements of highly sensitive cardiac troponin T (hs-cTnT), *N*-terminal pro B-type natriuretic peptide (NT-proBNP), highly sensitive *C*-reactive protein (hsCRP), and interleukin-6 (IL-6) were performed by electrochemiluminescence using standardized tests on an automated Roche Cobas 8000 clinical chemistry analyzer (Roche Diagnostics, Mannheim, Germany). Serum copeptin levels (pmol/L) were analyzed using immunologic trace technology (ThermoFisher/Brahms, Waltham, MA, USA).

### 2.4. Statistical Analysis

Baseline characteristics were compared using appropriate group comparison tests (e.g., the Mann–Whitney *u*-test for continuous parameters and the chi-square test for binary parameters). The impacts of classical risk factors and biomarkers on CAD groups were analyzed via multi-variable logistic regression models (function “glm” of the “R” statistical software package). We primarily considered the contrasts CAD0 + 1 vs. CAD2 (GC1: group comparison 1) and CAD0 vs. CAD1 + 2 (GC2: group comparison 2) as dependent variables of the models. Biomarkers were logarithmized prior to analysis and were considered as independent variables of the models. We included classical risk factors as further independent variables of the models to control for their effects. In detail, we included age, sex, log (BMI), smoking status, diabetes, log (LDL-C) adjusted for statin treatment, log (HDL-C), hypertension (defined as systolic blood pressure >140 mmHg, anamnestic information, or blood pressure medication), waist-to-hip ratio, log (triglycerides), and eGFR estimated with the CKD-Epi formula.

Receiver-operating characteristics (ROC) analysis was performed using the R pROC package and confidence intervals (CI). Comparisons between areas under the curve between ROCs were performed using Delong’s method. Survival analysis was performed with the “survival” package of “R.” Univariate comparisons between the groups were performed using the log-rank test. The impacts of scores on survival prediction were assessed via Cox-regression using the “coxph” package of R.

## 3. Result

### 3.1. Demographic Data and Clinical Characteristics of the Patient Cohort

There were 3072 patients included in this study, of which 1271 (41.4%) showed no coronary sclerosis (CAD0) and 631 (20.5%) showed non-obstructive coronary atherosclerosis (CAD1). Another 1170 patients (38.1%) showed significant coronary stenosis, ≥50%, of at least one coronary artery (CAD2). The demographic and laboratory characteristics for the CAD0, CAD1, and CAD2 groups are shown in Table 1.

A total of 3072 CAD patients were used for analysis (males *n* = 1966, 64%; females *n* = 1106, 36%). The age range of the total cohort was 25–87 years (62.2 ± 10.6). The table shows the medians (25–75 percentile) of the demographic data, the classical risk factors, and the selected biomarkers of the three CAD subcohorts stratified according to the stages of coronary artery stenosis CAD0 (no coronary sclerosis; 0–25% luminal reduction), CAD1 (25–50% luminal reduction), and CAD2 (≥one stenosis ≥ 50% luminal reduction).

### 3.2. Biomarkers Indicating the Angiographic Presence of Coronary Artery Disease

First, we studied the associations of the five serum biomarkers, hsTNT, NT-proBNP, copeptin, IL-6, and hsCRP, with the angiographic staging of CAD.

The biomarkers were compared among three predefined CAD severity subgroups (CAD0, CAD1, and CAD2). The median concentrations for all considered biomarkers across the CAD subgroups are shown in Table 1. For further analysis, we considered the differences in group comparison 1 (GC1: CAD0 + 1 vs. CAD2) and group comparison 2 (GC2: CAD0 vs. CAD1 + 2).

In the univariate analysis of GC1, hs-cTnT, NT-proBNP, and IL-6 differed significantly (hs-cTnT: Beta = 0.64, *p* = 7.1 × 10^−18^; NT-proBNP: Beta = 0.19, *p* = 1.9 × 10^−8^; IL-6: Beta = 0.29, *p* = 2.3 × 10^−6^; cardiovascular risk factors were adjusted).

After including all selected biomarkers and confounders in the multivariate GC1 model, hs-cTnT, NT-proBNP, and IL-6 remained significant. hs-cTnT showed the strongest effect for detecting patients with CAD 2 (hs-cTnT: Beta = 0.53, *p* = 2.0 × 10^−10^; NT-proBNP: Beta = 0.079, *p* = 0.049; IL6: Beta = 0.17, *p* = 0.030).

In the comparison group GC2 (CAD0 vs. CAD1 + 2), hs-cTnT, NT-proBNP, IL6, and hsCRP were univariately significantly associated with CAD 1 + 2 (hsTNT: Beta = 0.52, *p* = 2.0 × 10^−11^; NT-proBNP: Beta = 0.16, *p* = 1.6 × 10^−5^; IL6: Beta = 0.23, *p* = 5.2 × 10^−4^; hsCRP: Beta = 0.12, *p* = 5.2 × 10^−3^). In the multivariate analysis considering GC2 and including all selected biomarkers in the model, only hs-cTnT remained significant with a strong effect (Beta = 0.45, *p* = 1.9 × 10^−7^); copeptin reached nominal significance (Beta = −0.18, *p* = 0.014).

The multivariate effects of the conventional risk factors and selected biomarkers (Table 1) on GC2 are presented as a forest plot in Figure 1.

Since hs-cTnT represented a strong predictor, we next performed a ROC analysis for the prediction of significant CAD (GC1 and GC2). The hs-cTnT levels alone were indicative for GC1 and GC2 with an AUC of 0.69 (95% CI: 0.67–0.71), for both analyses (Appendix A). The classical risk factors age, sex, BMI, smoking, diabetes, LDL-C, HDL-C, hypertension, WHR, triglycerides, and eGFR without hs-cTnT showed AUCs of 0.76 (0.75–078) for GC1 and 0.76 (0.74–0.78) for GC2, respectively. The addition of hs-cTnT to the classical risk factors revealed significant but only small increments in the AUCs to 0.77 (95% CI: 0.75–0.79, *p* = 0.0013) and 0.77 (95% CI: 0.75–0.78, *p* = 0.029), respectively.

### 3.3. Selected Biomarkers Associated with Long-Term Mortality in Patients according to Coronary Artery Disease

To investigate the prognostic value of cardiac and inflammatory biomarkers for total mortality risk, we analyzed the selected coronary and inflammatory biomarkers in the context of CAD status and survival.

The 10-year survival probability for all participants was 80.1% (95% CI: 78.6%, 81.7%, data not shown). Long-term survival probability was significantly dependent on CAD status (*p* = 1.1 × 10^−23^ for global testing comparing the survival curves of the three CAD groups). The survival rates for CAD0, CAD1, and CAD2 after 10 years were 88.3% (95% CI 86.4–90.3%), 77.3% (73.8–80.9%), and 72.4% (69.6–72.3%), respectively (Figure 2).

Survival of patients with CAD0 vs. CAD1, CAD0 vs. CAD2, and CAD1 vs. CAD2 was significantly different, with *p*-values of 2.4 × 10^−10^, 1.2 × 10^−23^, and 0.018, respectively (Figure 2). Thus, survival rates of CAD1 and CAD2 were similar.

In the univariate survival analysis adjusting for classical risk factors in all CAD classes, all five selected biomarkers showed a significant association with total mortality (Table 2). In the multivariate analysis, only the hs-cTnT plasma levels remained an independent predictor of total mortality across all three CAD subgroups (CAD0: Beta = 0.44, *p* = 5.6 × 10^−3^; CAD1: Beta = 0.53, *p* = 2.2 × 10^−3^; CAD2: Beta = 0.39, *p* = 2.6 × 10^−4^). NT-proBNP remained significant only in CAD0 and CAD2 patients (CAD0: Beta = 0.14, *p* = 0.038; CAD2: Beta = 0.26, *p* = 1.7 × 10^−6^); and copeptin and IL6 remained significant in the CAD2 group only (copeptin: Beta = 0.34, *p* = 2.0 × 10^−4^; IL6: Beta = 0.27, *p* = 1.8 × 10^−3^).

All patients (*n* = 2935) in both CAD groups were analyzed by logistic regression models. Biomarkers were logarithmized prior to analysis. We adjusted for age, sex, log (BMI), smoking status, diabetes, log (LDL-C) adjusted for statin treatment, log (HDL-C), hypertension (defined as systolic blood pressure >140 mmHg, anamnestic information or blood pressure medication), waist-to-hip ratio, log (triglycerides), and eGFR estimated with the CKD-Epi formula.

Since survival rates in CAD1 and CAD2 were similar, we pooled the two groups for the following analysis. In the univariate analysis of the pooled group, all five biomarkers were significantly associated with survival. In the multivariate setting, i.e., combining all biomarkers, the biomarkers hs-cTnT (Beta = 0.42, *p* = 2.3 × 10^−6^), NT-proBNP (Beta = 0.20, *p* = 3.8 × 10^−6^), copeptin (Beta = 0.28, *p* = 2.0 × 10^−4^), and IL6 (Beta = 0.20, *p* = 5.9 × 10^−3^) remained significant. The CRP did not reach significance.

To assess the potential for improved risk stratification in patients with CAD1 and CAD2 and without CAD (CAD0), we analyzed the predictive value of classical risk factors in combination with the selected biomarkers.

In the model of classical risk factors, there was a pronounced survival difference between respective tertiles of the risk factor score (Figure 3). Ten-year survival rates for the three tertiles were 92.0% (89.7–94.4%), 78.2% (74.6–82.0%), and 51.5% (47.2–56.1%). Hazard rates with respect to the first tertile were 2.85 and 7.43; i.e., the survival of the third tertile was particularly dismal.

Adding hsTNT, NT-proBNP, copeptin, and IL6 to the risk score significantly improved the risk prediction for survival probability in CAD patients (*p* = 1.4 × 10^−22^, likelihood-ratio test). In the combined model of classical risk factors and biomarkers, survival rates for the three tertiles were 93.9% (91.8–96.1%), 81.4% (78.0–85.0%), and 46.8% (42.5–51.4%) (Figure 3). Hazard rates with respect to the first tertile were 3.13 and 11.2, respectively.

## 4. Discussion

To the best of our knowledge, this is the largest clinical study (3072 patients) with a standardized coronary angiographic assessment and a follow-up after more than 10 years to classify the roles of biomarkers in the context of coronary status and survival. The study was specifically designed to unravel the clinical value of non-invasive serum biomarkers to stratify CAD and long-term survival. The investigated biomarkers, hs-cTnT, NT-proBNP, copeptin, hsCRP, and IL-6, were specifically selected for the early and sensitive detection of patients at risk for CAD. They represent rapidly available, robust cardiac markers and highly sensitive inflammatory biomarkers that have already been linked to CAD.

We report two main findings. First, elevated serum levels of hs-cTnT allow for the stratification of patients with stable angina pectoris according to the stages of CAD with high precision, and the additional diagnostic value of NT-proBNP and IL-6 was only of secondary importance.

Second, hs-cTnT, NT-proBNP, IL-6, and copeptin are significant indicators of 10-year survival probability in high-risk patients with suspected CAD. The consideration of these biomarkers compared to classical risk factors significantly improves prognosis estimation.

Of all the studied biomarkers, hs-cTnT clearly showed the strongest ability to distinguish patients with different stages of CAD, and the other studied biomarkers were less important. Our findings are comparable to the previous results of the Heart and Soul study [12]. In this study, patients with stable CHD and elevated hs-cTnT levels showed multiple abnormalities in cardiac structure and function, which underlines the relevance of this biomarker.

The association of troponin with CAD has also been shown for troponin I, as reported by Beatty et al. [12]. Adamson et al. showed in the SCOT-HEART trial [13] in patients with suspected stable angina who underwent coronary computed tomography that higher cardiac troponin I levels were associated with obstructive CAD independent of known cardiovascular risk factors [13]. Our study and findings by others [6] clearly suggest the use of hs-cTnT as a clinical chemical signature for obstructive CAD, which may improve early diagnostic decision-making. Slightly elevated hs-cTnT levels should already be considered a sensitive indicator, even in the early stages of coronary disease. These patients could already be closely monitored and motivated to make lifestyle changes regardless of CHD stage. A very recent study reported by Mohebi, Jackson, et al. [14] showed patients without acute MI and CAD, but high concentrations of hs-cTnI, were associated with the presence of CAD and linked to increased risk of future CV events.

In our study, we also focused on prohormone NT-pro-BNP, which is released from the heart muscle in circulation in response to cardiomyocyte stretching and stress [15]. Today, NT-pro-BNP is an established biomarker for detecting ventricular dysfunction with prognostic value in heart failure. In our study, NT-pro-BNP was also associated with the stages of coronary disease. However, in combination with troponin and classical risk factors, it provides no additional diagnostic benefit in the stratification of CAD stages.

Another biomarker that is closely related to impaired cardiac function is provasopressin copeptin [16]. Copeptin is a small peptide in the *C*-terminal part of the pro-arginine vasopressin. It is directly released from the hypothalamus into the circulation caused by stress, blood pressure, and osmotic dysregulations [17,18]. It has been shown that copeptin can be used to predict acute myocardial infarctions accurately, while conventional troponin T is still undetectable (0 to 4 h) [19]. Little is known about the association between copeptin and stable CAD. In our study, copeptin provided no additional diagnostic benefit, which is analogous to NT-proBNP. Our finding is in contrast to a small study in 96 consecutive patients with documented CAD and chest pain, which suggested negative predictive value for copeptin levels, for excluding severe coronary stenosis [20].

One leading pathophysiological characterization of CAD is chronic inflammation and fibrotic proliferation of the arterial wall [21,22]. Several studies in the past have suggested associations of inflammatory biomarkers with atherosclerosis and potential anti-inflammatory treatments [4]. In our study, we investigated the associations of the inflammatory biomarkers IL-6 and CRP with the different stages of CAD. We found statistical associations; however, the differences were only minor and disappeared in the multivariate analyses. Earlier reports have demonstrated the significant impacts of inflammatory biomarkers on cardiovascular events [4,6]. However, in the context of classical risk factors, our study does not support earlier observations that CRP is a precise indicator of stable CAD [6,7]. This finding is in line with a Mendelian randomization study of the CRP genome, where genetic variations of the CRP gene causing elevated CRP levels were not associated with a higher degree of CAD or myocardial infarction [23]. Recent pathophysiological progress in the understanding of IL-6 signaling and the development of selective anti-IL-6 therapeutics in cardiovascular disease are being considered to prevent the progression of CAD. However, the associations between serum biomarkers of inflammation and stages of cardiovascular disease, as seen in our study, are only weak [24].

As a second goal of our study, we investigated the utility of the selected biomarkers for the prediction of 10-year survival in patients with suspected CAD. Interestingly, the survival rates of patients with CAD1 and CAD2 were almost the same. This result was unexpected. It can only be speculated that patients with advanced CAD receive more intensive medical and pharmacological treatments. To improve the conclusiveness of the analysis, we combined the two groups for survival analysis. In addition to the classical risk factors, hs-cTnT, NT-proBNP, copeptin, and IL6 significantly improved the risk prediction for survival in this combined group of patients.

Our findings regarding hs-cTnT and NT-proBNP levels are supported by results from the Ludwigshafen Risk and Cardiovascular Health (LURIC) study [25,26,27] and the AtheroGene cohort study [28]. However, the AtheroGene cohort study described CRP as an appropriate biomarker in this context, whereas in our observations, this was only true for the univariate analysis. In the multivariate analysis, CRP did not reach significance. We included the more sensitive inflammation marker IL-6 in our study, which was significantly associated with a worse long-term prognosis for patients. The results of our study can be seen in conjunction with the associations with CAD classes, and the above considerations of the possible options of anti-inflammatory therapy, and should be considered in follow-up studies. Recently, a 5-year study by Scicchitano P et al. [29] in 82 patients after carotid endarterectomy demonstrated soluble suppressor of tumorigenicity (sST-2) as a potential biomarker for atherosclerosis [30], as it was associated with long term-all cause mortality and symptomatic cerebrovascular events. The pathophysiology, specificity, and prognostic value of this interesting biomarker will be considered in our future study program.

Our study has some limitations. First, due to our study design, patients were recruited consecutively by elective admission for coronary angiography. Therefore, outpatient management before patient inclusion may differ significantly. Moreover, due to selection bias, our results cannot simply be extrapolated to the general population.

In addition, stress conditions and elevated blood pressure may have impacts on the plasma levels of biomarkers, such as copeptin and NT-proBNP. Patients with elevated blood pressure were found in all three groups, and blood pressure was included as a cofounder in the multivariate regression models. Patients with heart failure and reduced ejection fraction were not excluded, this could have affected the impact of the NT-proBNP levels. However, we wanted to analyze the data mimicking a “real world experience,” where patients may be submitted to an emergency room with suspected coronary heart disease with information about the EF not being readily available. Second, a large percentage of all three groups was treated with statins, which may have had an impact on the stress response of vessel wall inflammation in the atherosclerotic plaque and LDL cholesterol serum concentrations. Besides lipid-lowering medication, antihypertension medication and antidiabetic treatment were considered as relevant confounders. Anti-diabetic treatments, such as SGLT2 inhibitors, may influence NT-proBNP levels [31,32,33]. The effects of other medications are the subjects of current investigations. Finally, the individual causes of mortality of the patients were not available, and we were limited to reporting the associations of biomarkers with total mortality.

In summary, with the investigated cardiac and inflammatory biomarkers in this large cohort, we showed the potential for more precise biomarker-driven predictions of different stages of CAD and the risk of mortality. The most important biomarker for the different stages of CAD has been proven to be hs-cTnT [14]. For the prognosis of long-term survival, hs-cTnT, NT-proBNP, IL-6, and copeptin were all of importance. Further studies should follow to unravel the underlying pathophysiological mechanisms and define opportunities for future therapeutic interventions.

## Figures and Tables

**Figure 1 nutrients-14-03433-f001:**
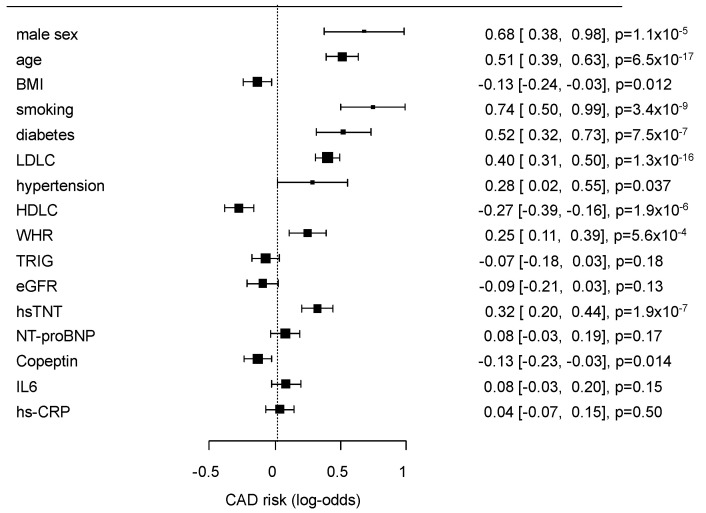
Results of multivariate analysis of classical risk factors and the selected biomarkers for the risk prediction of coronary artery disease (CAD0 vs. CAD1 + 2). The results are given as log-odds ratios, confidence intervals (CIs), and *p*-values.

**Figure 2 nutrients-14-03433-f002:**
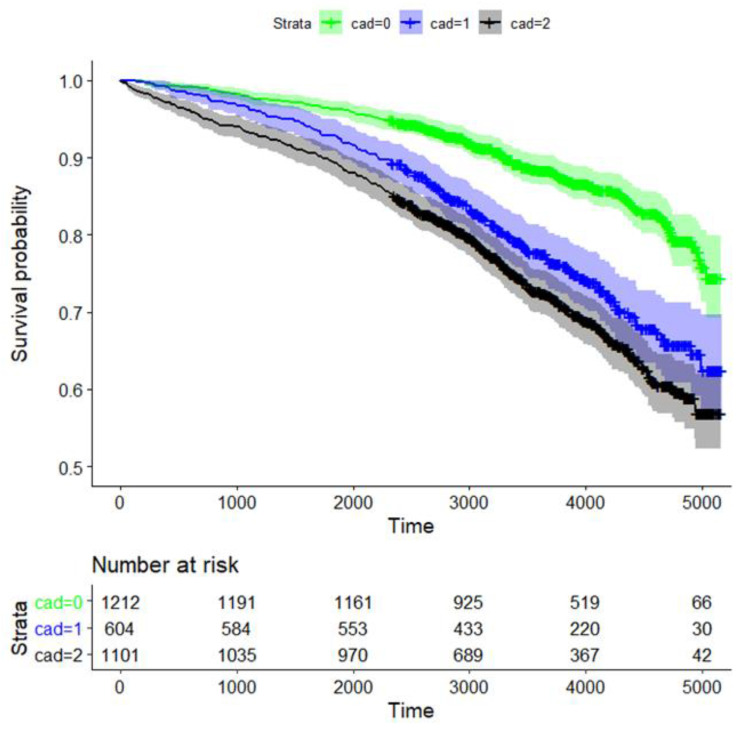
Kaplan–Meier curve analysis for survival of CAD patients based on the degree of coronary artery stenosis. Strata: CAD0 (green; no coronary sclerosis; 0–25% luminal reduction), CAD1 (blue; 25–50% luminal reduction), and obstructive CAD2 (black; ≥ one stenosis ≥50% luminal reduction). We also provide the time-dependent number of patients at risk.

**Figure 3 nutrients-14-03433-f003:**
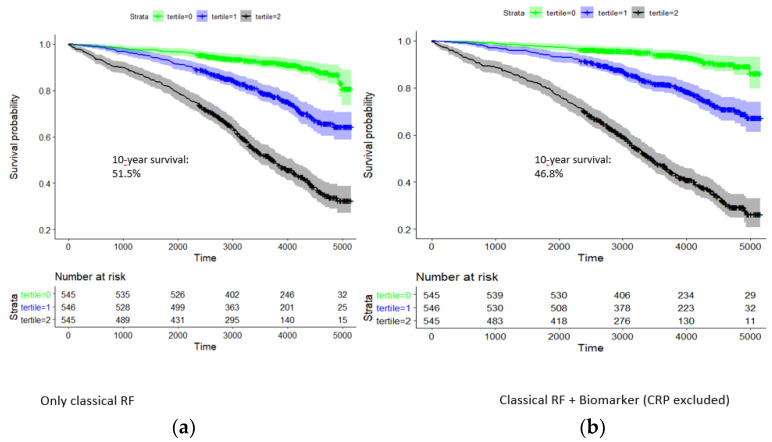
Kaplan–Meier curve analysis for 10-year survival of patients in the group CAD1 + CAD2: dependence on tertiles of a classical risk score and that combined with the biomarkers hsTNT, NT-proBNP, copeptin, and IL-6. Time is given in days. Strata: tertile 0 (green), tertile 1 (blue), and tertile 2 (black). (**a**) Classical risk factors: 10-year survival rate in highest tertile, 51.5%. (**b**) Classical risk factors and hs-cTnT, NT-proBNP, copeptin, and IL-6: 10-year survival in highest tertile, 46.8%.

**Table 1 nutrients-14-03433-t001:** Demographic data, classical risk factors, and the selected biomarkers of the studied CAD cohorts.

Demographic Data, Classical Risk Factors, and Selected Biomarkers of the Studied CAD Cohorts								
Characteristics	No CAD ^a^		CAD < 50		CAD ≥ 50		CAD0 + CAD1 vs. CAD2 (GC1) ^k^	CAD0 vs. CAD1 + CAD2 (GC2) ^l^
	Median	25–75 *p*	Median	25–75 *p*	Median	25–75 *p*		
*N* (%)	1271	41.4%	631	20.5%	1170	38.1%		
Sex (*N*)/Male	634	49.9%	419	66.4%	913	78.0%	<0.001	<0.001
Female	637	50.1%	212	33.6%	257	22.0%		
Age (years)	58.8	51.2–67.7	65.2	57.2–71.8	65.4	57.0–71.7	<0.001	<0.001
BMI ^b^ (kg/m²)	28.9	25.6–32.6	29.7	26.9–33.0	28.9	26.3–32.4	0.370	0.033
Waist-to-hip ratio	0.94	0.88–1.02	0.99	0.92–1.04	1.01	0.95–1.05	<0.001	<0.001
Diabetes ^c^ (*N*)	264	20.8%	221	35.0%	439	37.5%	<0.001	<0.001
Family history with MI ^d^ (*N*)	304	23.9%	138	21.9%	323	27.6%	0.007	0.289
Hypertension (*N*)	1040	81.8%	578	91.6%	1044	89.2%	0.002	<0.001
Antihypertension medication ^e^ (*N*)	1008	79.3%	562	89.1%	1017	86.9%	0.001	<0.001
Lipid lowering medication ^f^ (*N*)	359	28.2%	256	40.6%	504	43.1%	<0.001	<0.001
Current smoker (*N*)	212	16.7%	124	19.7%	275	23.5%	<0.001	0.001
Leukocytes	6.80	5.60–8.10	7.00	5.90–8.40	7.20	6.00–8.60	<0.001	<0.001
Total cholesterol (mmol/L)	5.40	4.73–6.14	5.29	4.54–6.17	5.47	4.62–6.36	0.057	0.890
LDL-cholesterol ^i^ (mmol/L)	3.24	2.60–3.88	3.20	2.58–3.89	3.43	2.66–4.23	<0.001	0.004
HDL-cholesterol ^j^ (mmol/L)	1.38	1.14–1.70	1.26	1.04–1.55	1.22	1.01–1.48	<0.001	<0.001
Triglycerides (mmol/L)	1.57	1.07–2.30	1.70	1.19–2.44	1.74	1.27–2.48	<0.001	<0.001
Creatinine (μmoL)	74.0	64.0–84.0	79.0	68.0–90.0	80.0	70.0–92.0	<0.001	<0.001
eGFR ^g^ (in mL/min/1.73 m²)	89.34	76.57–98.54	84.30	70.06–95.04	83.54	70.22–94.73	<0.001	<0.001
**Selected Biomarkers**								
Troponin T (pg/mL)	5.55	3.33–9.28	8.02	5.14–13.26	10.10	6.48–17.34	<0.001	<0.001
NT-pro BNP ^h^ (pg/mL)	98.6	46.0–218.2	132.1	60.8–353.4	173.5	75.5–463.8	<0.001	<0.001
CRP (mg/L)	1.83	0.97–3.87	2.39	1.16–4.77	2.41	1.14–5.16	<0.001	<0.001
Interleukin-6 (pg/mL)	2.01	1.50–3.69	2.67	1.55–4.54	3.04	1.73–5.91	<0.001	<0.001
Copeptin (pmol/L)	4.44	2.90–7.19	5.45	3.38–8.90	5.87	3.76–10.12	<0.001	<0.001

^a^ Coronary artery disease. ^b^ Body mass index. ^c^ Based on anamnestic information, medication with ATC-Code A10 or HbA1c > 6.5%. ^d^ Myocardial infarction. ^e^ Antihypertensive medication according to ATC-Code C02, C03, C07, C08, or C09. ^f^ Medication according to ATC-Code C10. ^g^ Glomerular filtration rate estimated according to Chronic Kidney Disease Epidemiology Collaboration (CKD-EPI) Creatinine. ^h^
*N*-terminal pro brain natriuretic peptide. ^i^ Low density lipoprotein cholesterol. ^j^ High density lipoprotein cholesterol. ^k^ GC1: group comparison 1. ^l^ GC2: group comparison 2.

**Table 2 nutrients-14-03433-t002:** Univariate and multivariate analysis of the selected biomarkers and the degree of coronary stenosis comparing the CAD0 vs. CAD1 + 2 groups (CG2).

	Univariate (All)				Multivariate (All)		
Biomarker	Beta	SE	Pval	Biomarker	Beta	SE	Pval
hsTNT	0.522	0.0772	1.39 × 10^−11^	hsTNT	0.453	0.0871	1.91 × 10^−7^
NT-proBNP	0.156	0.0362	1.57 × 10^−5^	NT-proBNP	0.0566	0.0416	0.173
Copeptin	−0.0933	0.069	0.176	Copeptin	−0.182	0.0739	0.0138
IL6	0.229	0.0659	0.000521	IL6	0.116	0.0805	0.151
hs-CRP	0.121	0.0432	0.00524	hs-CRP	0.0344	0.0516	0.505

Beta: Beta estimates of logistic regression, SE: standard error, pval: *p*-value.

## Data Availability

Not applicable.

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
