# Peer review of "Biomarkers for Non-Invasive Stratification of Coronary Artery Disease and Prognostic Impact on Long-Term Survival in Patients with Stable Coronary Heart Disease"

_nutrients, 2022, doi:10.3390/nu14163433_

Round 1

Reviewer 1 Report

The manuscript entitled “Biomarkers for Non-Invasive Stratification of Coronary Artery Disease and Prognostic Impact on Long-Term Survival in Patients with Stable Coronary Heart Disease.” was reviewed. This study attempted to investigate the association of the inflammatory biomarkers (copeptin, hsTNT, NT-proBNP, hsCRP, and interleukin-6), non-invasive identification of coronary artery disease patients, and mortality risk stratification. This manuscript is very interesting and well written. However, there are some issues that should be properly addressed.

1. The reviewer thinks the detection of an elevated cTnT value above the 99th percentile upper reference limit is defined as myocardial injury. [Circulation. 2018;138:e618-e651.] (Nonischemic myocardial injury may arise secondary to many cardiac conditions such as myocarditis, or may be associated with noncardiac conditions such as renal failure.) Thus, the authors should describe as “myocardial injury” but not as “acute coronary syndrome” in the Methods. (page 2, line 82) Further, should the patients with hs-cTnT ≥ 52 pg/ml be excluded in the present study?

2. Patients with severe systemic diseases, treated with immune-modulating therapies, and acute or chronic infectious diseases were excluded probably because of affecting the inflammatory biomarkers. Then, the reviewer thinks patients with heart failure reduced ejection fraction which have a great impact on the NT-proBNP should be also excluded.

3. If the present study aimed to investigate a risk stratification for cardiovascular prevention (page 2, line 63), the endpoints should include non-fatal acute myocardial infarction and cardiac death.

Author Response

  1. The reviewer thinks the detection of an elevated cTnT value above the 99th percentile upper reference limit is defined as myocardial injury. [Circulation. 2018;138:e618-e651.] (Nonischemic myocardial injury may arise secondary to many cardiac conditions such as myocarditis, or may be associated with noncardiac conditions such as renal failure.) Thus, the authors should describe as “myocardial injury” but not as “acute coronary syndrome” in the Methods. (page 2, line 82) Further, should the patients with hs-cTnT ≥ 52 pg/ml be excluded in the present study?

Reply:

We thank the reviewer for the helpful comments. We changed “acute coronary syndrome” in the methods section (page 2, line 84-85) to “myocardial injury” as suggested.

Patients with hs-cTnT ≥ 52 pg/ml were excluded in the present study to focus on patients with stable coronary heart disease patients. The present ESC guideline recommends this cut off using “Elecsys; Roche” for hs-cTn T analysis. (Reference 2: Collet, J.-P.; Thiele, H.; Barbato, E.; Barthélémy, O.; Bauersachs, J.; Bhatt, D.L.; Dendale, P.; Dorobantu, M.; Edvardsen, T.; Folliguet, T.; et al. 2020 ESC Guidelines for the management of acute coronary syndromes in patients presenting without persistent ST-segment elevation. Eur. Heart J. 2020, doi:10.1093/eurheartj/ehaa575).

For clarification,the following information was added in the method section (p 2, line 83-84):

“This cut off is based on the assay specific algorithm (Roche -Elecsys) for myocardial injury (2).”

  1. Patients with severe systemic diseases, treated with immune-modulating therapies, and acute or chronic infectious diseases were excluded probably because of affecting the inflammatory biomarkers. Then, the reviewer thinks patients with heart failure reduced ejection fraction which have a great impact on the NT-proBNP should be also excluded.

Reply:

Yes, as the reviewer suspected, we exclude known confounding factors of inflammation, to focus on the cardiac associated role of inflammatory biomarkers in stable coronary heart disease. We agree with the reviewer that patients with heart failure could have a great impact on the NT-proBNP levels.

Patients were primarily selected for stable coronary heart disease with the indication for coronary angiography. Patients with acute myocardial infarction and severe valvular disease were excluded. Patients with known history of heart failure were not excluded. We want to analyze the data mimicking “real world experience”, where patients may be submitted to an emergency room with suspected coronary heart disease and information about the EF is not readily available.

We did add a sentence in the discussion to point out this objective and this limitation to our study: Page 10, Line 377-381

“Patients with heart failure and reduced ejection fraction were not excluded, this could affect the impact on the NT-proBNP levels. This is a limitation of the study. However, we wanted to analyze the data mimicking “real world experience”, where patients may be submitted to an emergency room with suspected coronary heart disease and information about the EF is not readily available”.

  1. If the present study aimed to investigate a risk stratification for cardiovascular prevention (page 2, line 63), the endpoints should include non-fatal acute myocardial infarction and cardiac death.

Reply:

Thank you for your comment. In the present study we focus on the association of biomarkers with non-invasive identification of CAD patients and mortality risk stratification. The endpoints non-fatal acute myocardial infarction and cardiac death are included in the ongoing analysis of the life heart study. This important advice of the reviewer is part of our further research.

Reviewer 2 Report

This is good clinical research dedicated to inflammatory biomarkers in CAD with qualitative design and clear presentation. The manuscript may be accepted in present form.

Author Response

We thank the reviewer for the friendly comment . We appreciate your time and agreement  for our work. 

Reviewer 3 Report

To:

Editorial Board

Nutrients

Title: “Biomarkers for Non-Invasive Stratification of Coronary Artery Disease and Prognostic Impact on Long-Term Survival in Patients with Stable Coronary Heart Disease.”

Dear Editor,

I evaluated this paper and I think that:

-          Please include a flow chart of the study design.

-          What about other medications of patients? These might impact on results. Please describe medications and discuss such a point.

-          Authors should describe all the comorbidities of patients as they can impact on final results and on serum concentrations of biomarkers. Please discuss such a point.

-          A multivariate regression analysis should be performed in order to evaluate the impact of confounding factors on final results.

-          The role of sST2 should be better considered and discussed. Please discuss such a point in the Discussion section in relation to the papers from Scicchitano P et al. J Clin Med. 2022 May 31;11(11):3142 and Marzullo A et al. PLoS One. 2016 May 25;11(5):e0156315.

Author Response

Reviewer 3

Comments and Suggestions for Authors

Dear Editor,

I evaluated this paper and I think that: Please include a flow chart of the study design.

Reply:

We thank the reviewer for this suggestion. In the revised version of the manuscript, we added a flow chart of the study design (please see supplementary material 2 and information below). Page 2 line 89-90

Flow chart of the study design

Study population  N= 3,305 patients ( males N= 2,122 and females N= 1,183 ) admitted for clinically suspected chronic coronary syndrome, i.e., CAD and undergoing first-time coronary angiography.

Exclusion of 233 patients (of which 121 were excluded for unknown angiographic findings and 112 for hs-cTnT ≥ 52 pg/ml, indicating acute coronary syndrome).      

N= 3,072 patients (males N = 1,966, 64%; and females N = 1,106, 36%) were used for the analysis. The median follow up-time for survival analysis was 10 years (range 0-14).

Groups studied:

CAD0 (no coronary sclerosis),

CAD1 (non-obstructive coronary sclerosis, i.e., plaques with < 50% luminal reduction),

obstructive CAD2 (≥ one stenosis ≥ 50% luminal reduction).

We primarily compared:

CAD0 + 1 vs. CAD2 (GC1: group comparison 1),

CAD0 vs. CAD1 + 2 (GC2: group comparison 2).

  • What about other medications of patients? These might impact on results. Please describe medications and discuss such a point.

Reply:

Medications were documented according to the ATC-Codes. Antihypertension medication, lipid lowering medication, antidiabetic treatment were considered as relevant confounders. For example, statins have known anti-inflammatory effects and anti-diabetic treatment as SGLT2 inhibitors may influence NT-proBNP.  We agree with the reviewer's comment that the influence of other medications should also be analyzed in more detail. However, this was not part of the concept of the present study, but will be done in later analyses.

We added the following sentence in the method section, page 3, line 101:

“Medications were documented according to the ATC-Codes.”

We added the following sentence discussion, page 11, line 385-388:

“Beside lipid lowering medication, also antihypertension medication, antidiabetic treatment were considered as relevant confounders. Anti-diabetic treatment as SGLT2 inhibitors may influence NT-proBNP levels. The effects of other medications are the subject of current investigations “

 References (C,D,E)

  • Authors should describe all the comorbidities of patients as they can impact on final results and on serum concentrations of biomarkers. Please discuss such a point.

  Reply:

We did add a sentence for clarification in the discussion, page 3 , line 96-100

“Frequent comorbidities were considered in the evaluation as confounding factors: diabetes mellitus, obesity, hyperlipidemia, hypertension, impaired renal function. These comorbidities have a particular impact on metabolic parameters, inflammatory markers, hormonal regulations, and the half-life of circulating biomarkers (e.g., renal function). Other comorbidities were documented and are the subject of ongoing investigations”. 

  • A multivariate regression analysis should be performed in order to evaluate the impact of confounding factors on final results.

Reply :

      We agree with the reviewer. Actually, we performed multi-variable regression analysis controlling for classical risk factors of coronary-artery disease. We apologize that this was not stated clearly enough in the previous version of the manuscript.

We rewrote the respective paragraph in the methods section, page 3, lines 128 to 138

      “The impact of classical risk factors and biomarkers on CAD groups was analyzed   by multi-variable   logistic regression models (function “glm”   of   the   “R”   statistical   software
package). We primarily considered the contrasts CAD0 + 1 vs. CAD2 (GC1: group comparison 1) and CAD0 vs. CAD1 + 2 (GC2: group comparison 2) as dependent variables of the models. Biomarkers were logarithmized prior to analysis and were considered as independent variables of the models. We
included classical risk factors as further independent variables of the models to control for their effects. In detail, we included age, sex, log (BMI), smoking status, diabetes, log (LDL-C) adjusted   for   statin   treatment, log   (HDL-C),   hypertension (defined as systolic blood pressure > 140 mmHg, anamnestic
information, or blood pressure medication), waist-to-hip ratio, log (triglycerides), and eGFR estimated with the CKD-Epi formula. “

  • The role of sST2 should be better considered and discussed. Please discuss such a point in the Discussion section in relation to the papers from Scicchitano P et al. J Clin Med. 2022 May 31;11(11):3142 and Marzullo A et al. PLoS One. 2016 May 25;11(5):e0156315.

Reply:

We thank the reviewer for the recent sST2 publication. We did add a short comment in the discussion. Page 10, line 361- 365:

“Recently, a 5 years study in 82 patients after carotid endarterectomy demonstrated soluble suppressor of tumorigenicity (sST-2) as a potential biomarker for atherosclerosis associated with long term-all cause mortality and symptomatic cerebrovascular events. Pathophysiology, specificity and prognostic value of this interesting biomarker will be considered in our future study program.”

Reference.(A, B)
